



# A long-term hydrometeorological dataset (1993-2014) of a northern mountain basin: Wolf Creek Research Basin, Yukon Territory, Canada

Kabir Rasouli[1, 2], John W. Pomeroy[2], J. Richard Janowicz[2, 3*], Tyler J. Williams[2, 3], and Sean K. Carey[4]

[1]Department of Geoscience, University of Calgary, Calgary, AB, Canada

[2]Centre for Hydrology, University of Saskatchewan, Saskatoon, SK, Canada

[3]Environment Yukon, Water Resources Branch, Whitehorse, YT, Canada

[4]School of Geography and Earth Sciences, McMaster University, Hamilton, ON, Canada

*Correspondence*: Kabir Rasouli (kabir.rasouli@usask.ca)

[*]Deceased.

**Abstract.** A set of hydrometeorological data is presented in this paper, which can be used to characterize the hydrometeorology and climate of a subarctic mountain basin and has proven particularly useful for forcing hydrological models and assessing their performance in capturing hydrological processes in subarctic alpine environments. The forcing dataset includes daily precipitation, hourly air temperature, humidity, wind, solar and net radiation, soil temperature, and geographical information system data. The model performance assessment data include snow depth and snow water equivalent, streamflow, soil moisture, and water level in a groundwater well. This dataset was recorded at different elevation bands in Wolf Creek Research Basin, near Whitehorse, Yukon Territory, Canada representing forest, shrub tundra, and alpine tundra biomes from 1993 through 2014. Measurements continue through 2018 and are planned for the future at this basin and will be updated to the data website. The database presented and described in this article is available for download at https://doi.org/10.20383/101.0113.

## 1 Introduction

The availability of hydrometeorological data is limited in northern latitudes in general and in northern mountains in particular. This is because of a sparse monitoring network, harsh weather, and high cost of experiments and instrument maintenance in these environments (Klemeš, 1990). The number of stations that record a complete hydrometeorological dataset in the northern latitudes is limited and declining (Laudon et al., 2017). Wolf Creek Research Basin (WCRB) was established in 1992 to carry out water and climate research and is located in the Yukon



Territory, Canada (≈61ºN). This basin is operated by Environment Yukon, University of Saskatchewan and McMaster University with support from the Global Water Futures program and provides a long-term dataset for precipitation, air temperature, humidity, wind, radiation, soil moisture, soil temperature, streamflow, and snowpacks at multiple elevations. WCRB includes meteorological stations located from low to high elevation that include a special emphasis

on measuring snowpack and snowfall, and discharge gauges to measure streamflow in the main outlet and several tributary streams.

The diversity of the basin, combined with the available long term comprehensive hydrometeorological data, is responsible for the popularity of WCRB as a site to carry out cold regions research by scientists from across Canada

and abroad. Road access in summer, local accommodation, a major airport and upper air station in Whitehorse, simple winter logistics, data availability and ecological diversity make WCRB an ideal location for remote sensing validation and snow modelling activities. WCRB can be used for detailed modelling studies as it has been densely monitored, has sufficiently long observation records and extensive parameter measurements (Pomeroy et al. 1999; Pomeroy et al. 2003, 2006; McCartney et al. 2006; Carey et al. 2007; Dornes et al. 2008; Quinton and Carey 2008; MacDonald et al.

2009; Rasouli et al. 2014, 2018; Rasouli, 2017).

## 2   Site Description

WCRB is a subarctic headwater basin (Figure 1) with a long cold snow season characterized by low precipitation

(Figure 2). The drainage area in this basin is 179 km$^2$ and elevation ranges between 660 and 2080 m above sea level. The basin is composed of four monitored sub-basins of Upper Wolf Creek, Coal Lake, Granger Creek, and Lower Wolf Creek, one non-monitored sub-basin of Middle Wolf Creek, and also three distinct ecological biomes; alpine tundra (20%), shrub tundra and taiga (58%), and boreal forest (22%) (Figure 1). At the very highest elevations, bare rock, short tundra mosses, and grasses dominate land cover. Above the treeline, shrub tundra with dwarf birch and

willow shrub heights from 30 cm to 3 m occupies the majority of the basin (Jorgenson and Heiner, 2004). The taiga at middle to lower elevations is dominated by shrubs and sparse black spruce woodland. At the lowest elevations are lodgepole pine, white spruce, and trembling aspen forest stands (Francis et al. 1998). Lewkowicz and Ednie (2004) estimate that 43% of WCRB contains permafrost. The basin physiography is described in detail by Rasouli et al. (2014). Meteorological measurements are taken at three primary stations in each of the biomes (Table 1; Figure 1)

and are described as:

(1)   **Alpine tundra station.** A windswept, high alpine tundra plateau along the drainage divide at the northern edge just outside of the basin. Vegetation is sparse consisting of moss and lichens with scattered boulders up to 1 m tall. Sensors are mounted on a 3 m tower. Additional snow sensors are Geonor T200B and Nipher snowfall gauge.



(2) **Shrub tundra – taiga station.** An east-facing, moderate hillslope consisting of undulating terrain. Vegetation is tall shrubs (1-3 m) with sparsely scattered spruce (spaced about 100 m apart). The presence of strong winds yet restriction on snow redistribution by shrubs makes this an ideal site for snowfall wind undercatch studies (Pan et al., 2016; Kochendorfer et al., 2017). Sensors are mounted on a 5.2 m triangular tower. Additional snow sensors

are Ott Pluvio version 200 and version 400, Geonor T200B, BC Standpipe precipitation gauge, and Nipher snowfall gauge.

(3) **White spruce forest station.** A dense canopy of mature white spruce forest (12-18 m) near the Wolf Creek basin outlet. Well suited for assessing the effects of forest cover on snow water equivalent (SWE). Sensors are mounted on a 21 m triangular tower at various heights below, in and above the forest canopy. The tower at this site was

relocated in 1996 and caution should be used comparing data pre- and post-move. Additional snow sensors: Geonor T200B, Nipher Snowfall gauge.

### 3   GIS Drainage Basin Descriptors

A digital elevation model, DEM with 30 m cell resolution was prepared by Dr. Lawrence Martz (Dept of Geography, University of Saskatchewan). DEM, geographic information system files for streamflow gauges, meteorological stations, groundwater well, boundaries of the drainage basin and sub-basins, streamflow tributaries, and lakes are provided (Figure 1). An image of land-cover is also presented.

### 4   Data description

Hydrometeorological data are available for WCRB for water years (WY) from 1993-1994 to 2013-2014 for the three main meteorological stations, one in each primary biome, and four streamflow gauges (Figure 1; Table 1). Measurements of snow depth and density along snow survey transects were collected at least monthly by Environment

Yukon and university researchers at each of the three meteorological stations. These measurements provide model diagnostic information in each biome (Pomeroy and Granger, 1999). High quality meteorological measurements of hourly air temperature, relative humidity, rainfall, wind speed, incoming and outgoing shortwave radiation, net radiation, soil heat flux, soil moisture and temperature, and daily precipitation observations for three stations (Table 2) and hydrological measurements of hourly streamflow data recorded by four gauges, snow water equivalent, snow

depth, and water level at one groundwater well (Table 3) are presented for the period of 1993-2014 in this paper. Table 2 and 3 list the sensors measuring meteorology, streamflow, soil parameters, groundwater, and snow in the three sites in WCRB and provides the height or depth of the instruments, long-term climatological averages, and periods of observation record. All stations record data at 30 minute intervals, but are reported hourly (except for precipitation data that is reported daily) in this paper with continuous records for several parameters at the main stations. All stations



use Campbell Scientific dataloggers and data have been transmitted using cellular modems and spread-spectrum radios since 2016.

Precipitation has been measured by tipping bucket rain gauges, unshielded BC-style Standpipe precipitation gauges,
Nipher-shielded Meteorological Service of Canada (MSC) snowfall, and recently Alter-shielded Ott Pluvio and Geonor gauges. Ott Pluvio (since 2013) and Standpipe gauges were installed only at the shrub tundra site. Geonor gauges were installed in 2007 at the forest and in 2010 at the shrub tundra and alpine sites. The Campbell SR-50 ultrasonic snow depth sensors can contain substantial noise which is in part due to vegetation, falling snow particles, and movement of the sensor. As mentioned, the forest tower relocated in June 1996 so the first few years of data are
similar but not directly comparable to the new site. Snow surveys have been carried out once a month and snow depth, density, snow water equivalent (SWE) data have been collected over the period of 1994-2014 at three traditional snow survey sites at the alpine, and shrub tundra, and forest sites. Each has a 25 point transect with 5 density measurements.

Water level at streams was monitored continuously from ice thaw in early May to freeze-up in early October.
Discharge measurements involving manual stream velocity measurements throughout the year are taken approximately once a month at Upper Wolf Creek, Coal Lake outlet, and Wolf Creek at Alaska Highway. Solinst leveloggers are currently used for measuring level at the four hydrometric stations (Figure 1). Other instruments that were previously used for measuring water level and streamflow are HOBO U20-001-04 pressure transducers, Ott Thalimedes shaft encoders and Leupold and Stevens A-71 Strip Charts. The Alaska Highway hydrometric station has
the most field visits due to ease of access, has received the most attention due to its importance as the basin outlet, and has bridge access for the measurement of exceptionally high flows. As a result, it has the highest quality discharge observations in the basin. However, at very high flows, the channel is not ideal, particularly on the left bank where a small stream enters the creek adjacent to the gauging well. Additionally, there is uncertainty with the staff gauge reading at high flows, due to hydraulic jumps. At the highest of flows, it may be best to rely on the instrument gauge
readings. The Coal Lake station is located in a location with good channel geometry for rating development, yet has been challenging to instrument early to capture high flows. Historically, at least one and possibly more outburst flood events have occurred which produce high discharge early in the season. At the very highest water levels experienced at the station, the site is susceptible to flooding. Granger Creek is a steep, narrow, rough, and turbulent creek that is not ideal for velocity-area measurements or rating curve development, and so there is likely considerably error
associated with the site. A very high water discharge measurement has never been made at the site, and benchmarks have only been in use since 2012 so water level data is not transferable from older data. The channel is well controlled so that flooding is not an issue. The flashy nature of high water at Granger Creek also makes it difficult to observe snowmelt induced peak flows, which generally occur towards late in the day. The Upper Wolf Creek station was partially buried by a debris flow caused by an intensive convective storm and flood in the summer of 2013 and has
been inactive since then. The site has historically been a source of difficulty as it has already been relocated once due



to beaver activity. It is difficult to find an adequate control at Upper Wolf, with high flows often overtopping the bank and making it difficult to measure high water.

5 Volumetric soil moisture (unfrozen water content) and soil temperature were recorded every 30 min over the period of 1996-2014 at the meteorological stations in soil profiles at 0.05, 0.15, 0.30, 0.80 m depths (Table 3). A groundwater well was installed in 2001, approximately 14 km south of downtown Whitehorse, Yukon Territory (Figure 1). It is an open-hole observation well on bedrock with 48.8 m depth and 15.24 cm diameter and an estimated production capacity of 0.6 L/s. Water level depth and temperature have been monitored hourly since 2001 at this well.

## 5  Data processing and adjustments

Data loss has been a common problem during the early winter due to power failure at stations with the onset of darkness and cold weather. The Standpipe gauge at the shrub tundra site is a pressure sensing precipitation gauge that broadcasts real-time data every 3 hours. Nipher-gauge solid precipitation measurements were corrected using a wind

undercatch correction equation (Goodison et al., 1998) with wind speeds measured from nearby gauge-height anemometers. Precipitation data with poor quality were removed and gaps in data were filled by establishing regression equations for meteorological variables between each of the three meteorological stations and the Whitehorse WSO station, which is located 13 km from WCRB. The equations to fill the gaps in precipitation and to correct for the wind undercatch were listed in Table 4. Precipitation data from the Whitehorse Airport and Whitehorse

Riverdale stations were taken to fill the gaps primarily in WY of 1996-1997. Changes in elevation rather than distance between stations are assumed to be responsible for the meteorological variation in this study basin. First, total daily precipitation was separated to snowfall and rainfall phases using Harder et al., (2013). Then, the snowfall proportion of precipitation was corrected for wind undercatch in each site. When data was missing, precipitation at the shrub tundra or alpine station was estimated solely based on the Whitehorse WSO measurements. Whitehorse WSO snowfall

measurements are collected using a Nipher and require appropriate wind correction using WSO wind speed measurements. Whitehorse WSO precipitation was assumed equal to precipitation at the forest site because of their proximity and similar elevations. Using available monthly totals, a relationship was found between cumulative snowfall at the Standpipe ($S_S^{sh}$) and the Nipher gauges to calculate "Nipher equivalent" snowfall ($S_N^{sh}$) at the shrub tundra site as in Eq. (1). When Standpipe data were not available, a relationship between Nipher equivalent cumulative

precipitation at the shrub tundra site and the Nipher precipitation at the Whitehorse WSO site ($S_N^{WH}$) was used as in Eq. (2). The snowfall data at the shrub tundra site also require a Nipher wind undercatch correction. To estimate missing rainfall data in the shrub tundra site and the Standpipe gauge, a relationship between cumulative rainfall at Standpipe gauge ($R_S^{sh}$) and rainfall at Whitehorse WSO site was used as in Eq. (3). To estimate missing snowfall at the alpine site a relationship was used for uncorrected cumulative snowfalls between the alpine and shrub tundra



Nipher gauges based on available monthly data as in Eq. (4). This equations can also be applied to the calculated daily "Nipher shrub tundra equivalent" in Eq. (1). To estimate missing rainfall data at the alpine site the relationship between cumulative rainfall at the alpine Geonor ($R_G^A$) and the shrub tundra Standpipe (Eq. 5) was applied to the daily rainfall data calculated or measured for shrub tundra as in Eq. (3). A relationship between cumulative rainfall data recorded

at the shrub tundra Standpipe and Geonor was found as in Eq. (6).

For undercatch correction, gaps in wind speed data (U) also were filled by interpreting from other stations based on best fit linear equations. The catch ratio (CR) for the Nipher gauges was calculated from Eq. (7), in which wind speed in m s$^{-1}$ is measured at the Nipher gauge height (Goodison et al., 1998). In order to calculate wind speed at gauge

height, roughness lengths of 0.001 m at the alpine and 0.1 m at the shrub tundra were used. Regression equations were used to fill the gaps in other meteorological variables in the three stations representing forest, shrub tundra, and alpine biomes in WCRB. The regression equations for air temperature, relative humidity, and wind speed were provided in Table 5. Air temperature data has undergone a thorough quality control process. This included comparing air temperature among the three sites to look for unrealistic differences. Where available, air temperatures were also

compared between multiple measurements made at the same site. The gaps in hourly air temperature data that were between 6-13% of the total data were also filled by regressions established between hourly temperatures in the three sites in WCRB (Table 5). When data is missing for a few hours, an interpolation between air temperatures before and after the gap is used.

The relative humidity (*RH*) data at the alpine site had periods of *RH* > 100% prior to correction. For the first few years of the archive, the data often exceeded 100. From 1995-2005 the peak *RH* being recorded gradually increased each year. Starting in September 2005, the sensor was replaced with a new version, and data quality improved, rarely exceeding 100%. From 2010 to the present time the data was almost entirely below 100%. For a few years between 1998 and 2000, there was also a second *RH* sensor at the alpine site. This sensor confirmed that the main tower *RH*

sensor was in fact providing values that were too high, but also showed that for low *RH* values it was also reporting values that were too low. As such, an equation was developed to reduce the variation of the raw data ($RH_{raw}$) using a second order polynomial as in Eq. (8) in which $a$ and $b$ are constant values (Table 4). The *RH* data at the shrub tundra and forest sites, which have similar patterns to the alpine site, were quality-controlled and unreasonable data were corrected.

The REBS instrument net radiation (Rn) measurements require wind speed corrections and are now understood to have considerable measurement uncertainty. REBS were operation prior to 2006. When wind speed (U) data was not available a generic equation as in Eq. (9) was applied and when wind speed data was available Eq. (10) is used (Table 4). In all cases, the propeller anemometers were the first choice used for wind speed, if not available, cup anemometers

were used. The NR-Lite2 instruments also required a post correction in some cases. This correction is only required



when wind speed exceed 5 m s$^{-1}$ (Eq. 11). After the installation of the CR1000 dataloggers in 2013, this correction equation for the NR-Lite2 was written directly into the program and therefore post-corrections are no longer required. The original data were not destroyed with any of these corrections, so "uncorrected" net radiation data prior to wind corrections is available for all sites. Shortwave radiation data was collected initially by LiCor Li200S and after 2005

5    by Kipp-Zonen CMP3-L sensors; these data needed to be cleaned to change night-time values to zero, but in other cases the data still has small non-zero values. The CR1000 program now automatically changes negative radiation values to zero, but there can still be small erroneous positive values. Snow or hoarfrost can frequently build up on the solar radiation sensors in mid-winter when wind speeds are low. It is important to keep detailed field notes about icing to assist with identifying uncertainties with this data.

Snow depth data from the Campbell Scientific Canada SR50 ultrasonic depth sounder are adjusted to compensate for the effect of temperature on the speed of sound and require a further quality control to smooth out half-hour data and remove the most obviously erroneous data due to precipitation, blowing snow and movement of the gauge or growth of vegetation under the gauge (vegetation height can be misinterpreted as snow depth). Data was shifted up or down

to ensure that the snow depth started and ended the snowcovered period at 0, and albedo increases were used to help identify the snowcovered periods. These corrections have been applied to the entire historical record for all sites and have produced useable datasets at a daily time scale. The snow depth sites, however, have occasionally been disturbed in the past by people standing too close to or falling under the sensor, which adds uncertainty.

Corrections were required to account for differences in soil moisture data over time, some of which are likely due to

changes in dataloggers or programs. There was a shift in soil moisture values at the forest site after 2005 at depths of 0.3 m and 0.8 m below ground surface. This shift was corrected and the data recorded before 2005 was corrected to match with the range of the data recorded after 2005. Neither the shrub tundra nor alpine pits have had this correction.

**6    Monthly climatological averages**

The long-term monthly averages are presented over the 22-year period in this section. Precipitation reaches maximum 59 mm at the shrub tundra site in July and minimum 8 mm in April at the forest site (Figure 2). Precipitation maxima fall as rain in summer months.  The rainfall ratio, which is defined as the ratio of annual rainfall to total annual precipitation decreases with elevation: 0.61 at the forest site (750 m elevation), 0.56 at the shrub tundra site (1250 m

elevation), and 0.51 at the alpine site (1615 m elevation). The long-term climatological averages show that air temperature is below the freezing-point for seven months at all three sites in the WCRB. The inter-monthly variation of air temperature is the largest in the forest site which reaches -18°C in January and 13.6°C in July (Figure 2).  The relative humidity is the lowest in April and May at the forest site and the highest in October at the alpine site in WCRB (Figure 3). The recorded wind speed increases with elevation and reaches the maximum value of 35 m s$^{-1}$ at the alpine





site in WCRB over 1993-2014. The prevailing wind direction is southeasterly in the alpine site and above canopy in the forest site and southwesterly in the shrub tundra site (Figure 4). Maximum wind speed reaches 8 m s$^{-1}$ above canopy in the forest site (Figure 4c), which is the lowest amongst the three sites. The second prevailing wind direction is northwesterly in all three sites in WCRB (Figure 4). Figure 5 shows the annual cycle of streamflow at Alaska

Highway and Coal Lake gauges. Streamflow records show the lowest rates in March and then rises sharply in both gauges and reaches their peak values in July. Monthly patterns of streamflow at Coal Lake as a sub-basin and at Alaska Highway as outlet of the WCRB are almost the same. Figure 5 shows the annual cycle of water level in the groundwater well. On average, the water level drops to 16.2 m below ground surface in April and rises to its maximum level, 15.87 m below ground surface in August and September. Figure 6a shows annual cycle of the soil unfrozen water content at

the shrub tundra site at top 1 m soil depth. Soils are dry in winter and soil unfrozen water content is below 0.15 in all soil layers at soil profile with 1 m depth. Soil moisture increases rapidly in April and reaches its maximum value of 0.5 m$^3$ m$^{-3}$ in May at top 0.2 m soil depth. Soils freeze from November to April down to 0.7 m below ground surface and soil temperature reaches 8 in July and August (Figure 6b). Soil temperature at top 0.20 m soil layer varies dramatically with air temperature seasonality. Soil temporaries at lower layers show a delay in response to air

temperature variations.

## 7  Example of  data use

Data over the period of 1993-2011 were used by Rasouli et al. (2014, 2018) and Rasouli (2017) to investigate the

20 change in mountain snow and runoff regimes under simulated warmer and wetter conditions in an uncertainty framework. Future projections from eleven of the regional climate models for the A2 scenario for the period 2041-2070 were used to perturb records of observations of temperature and precipitation in WCRB.

## 8  Availability of the dataset

The database presented and described in this article is available for download from Federated Research Data Repository at https://doi.org/10.20383/101.0113. Data at the different stations are provided in separated files in csv format. The DEM of the study area (179 km²) at 30 m resolution is also provided in NAD_1983_UTM_Zone_10N coordinates. New data will be added in the database on a yearly basis and made available to the community.





## 9 Final remarks

The data from the Wolf Creek Research Basin have contributed significantly to our understanding of snow and runoff processes in subarctic mountainous environments. The long-term dataset at multiple elevations and land-cover types can be used to examine the variability of hydrometeorological fluxes with elevation and land-cover. As an example,
the data have been used to investigate the hydrological influence exerted by shrub tundra (Pomeroy et al., 2006). The unique dataset will be valuable to research communities working on mountain and northern hydrology for various purposes such as hydrological model development, assessment of climate change impacts, and inter-site comparison of hydrological processes.

Acknowledgments

The authors wish to acknowledge three decades of Wolf Creek Research Basin operation with substantial contributions to installation, maintenance and operation of the core snow surveys, meteorological stations and hydrometric stations under inclement conditions from Glenn Ford, Glen Carpenter, Kerry Paslowski, Martin Jasek, Dell Bayne, Newell Hedstrom, Raoul Granger, Cuyler Onclin, Michael Solohub and Richard Essery amongst many others. Funding for
the basin has come from Yukon Environment, Environment Canada, DIAND, NSERC, CFCAS, NERC, Welsh Assembly Government, NOAA, and other sources. The research was funded by the NSERC's Discovery Grants and Post-doctoral Fellowship, Changing Cold Regions Network, Alexander Graham Bell Canada Graduate Scholarship-Doctoral Program and Global Water Futures.

Author contributions. Kabir Rasouli and Tyler J. Williams cleaned, organized, and corrected the data and wrote the first draft of the paper, John W. Pomeroy, J. Richard Janowicz and Sean K. Cary designed and built the instrumented research basin, collected data, managed the data collection over the last three decades and contributed to the writing of the manuscript.

Competing interests. The authors declare that they have no conflict of interest.

Disclaimer. Any reference to specific equipment types or manufacturers is for informational purposes and does not represent a product endorsement.

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

25



**Table 1. Description of the main meteorological and hydrometric stations within Wolf Creek Research Basin**

| Station | Elevation [m] | Latitude [N] | Longitude [W] | Site details |
|---|---|---|---|---|
| (1) Meteorological station | | | | |
| Shrub tundra | 1250 | 60∘31.34′ | 135∘11.84′ | east facing moderate slope |
| Alpine | 1615 | 60∘34.04′ | 135∘08.98′ | windswept ridge top plateau |
| Forest | 750 | 60∘35.76′ | 134∘57.17′ | gently undulating terrain |
| Whitehorse WSO | 706 | 60∘44.00′ | 135∘05.00′ | at Environment Canada site |
| (2) Hydrometric station | | | | |
| Upper Wolf Creek | 1295 | 60∘29.45′ | 135∘17.50′ | drainage area: 14.4 km² |
| Coal Lake Outlet | 1190 | 60∘30.61′ | 135∘09.74′ | drainage area: 70.5 km² |
| Granger Creek | 1312 | 60∘32.79′ | 135∘11.08′ | drainage area: 7.6 km² |
| Alaska Highway | 703 | 60∘36.00′ | 134∘57.00′ | drainage area: 179 km² |
| (3) Groundwater well | | | | |
| Wolf Creek Well | 750 | 60∘34.41′ | 134∘57.7′ | water level and temperatures |



**Table 2. Meteorological data measured in three site in the basin including variables, climatological water year mean values, current sensors, and measurement height**

| Variable | Site | Sensor | Height/ depth[m] | Mean | Record period |
|---|---|---|---|---|---|
| Precipitation [mm] | Alpine | Texas TE525M TBRG Geonor T200B | 1.87 | 384 | 1993-2014 daily |
| | Shrub tundra | Texas TE525M TBRG Standpipe Ott Pluvio version 200 Geonor T200B | 4.76 | 403 | 1993-2014 daily |
| | Forest | Texas TE525M TBRG Geonor T200B | 21.34 1.75 | 278 | 1993-2014 daily |
| Air temperature [°C]& humidity [%] | Alpine | Vaisala HMP35CF HM45C212 (post-2005) | 2.05 2.05 | -3.5 & 78 | 1993-2014 hourly |
| | Shrub tundra | Vaisala HMP35CF HMP45-212(post-2005) | 2.6 2.6 | -1.9& 77 | 1993-2014 hourly |
| | Forest | Vaisala HMP35CF HMP45-212(post-2007) | 21.34 21.34 | -1.4& 75 | 1993-2014 hourly |
| Wind speed [m s$^{-1}$] & direction [-] | Alpine | NRG 40 Cup Anemometer NRG 200P | 2.75 2.75 | 3.6 (SE) | 1993-2014 hourly |
| | Shrub tundra | NRG 40 Cup Anemometer RM Young 5103 | 4.6 4.98 | 1.8 (SW) | 1993-2014 hourly |
| | Forest | NRG 40 Cup Anemometer RM Young 5103 | 19.82 20 | 1.4 (SE) | 1993-2014 hourly |
| Net radiation [Wm$^{-2}$] | Alpine | REBS Q6.0 NR-Lite (post 2004) | 1.68 1.68 | 40 | 1993-2014 hourly |
| | Shrub tundra | REBS Q6.0 NR-Lite (post 2006) | 3.10 3.18 | 61 | 1993-2014 hourly |
| | Forest | REBS Q6.0 NR-Lite (post 2006) | 19.82 19.82 | Above canopy,77 | 1993-2014 hourly |
| Solar radiation [Wm$^{-2}$] | Alpine | LiCor Li200S Kipp & Zonen CMP3-L (post 2005) | in: 2.93 out: 1.69 | in: 231 out: 95 | 1993-2014 hourly |
| | Shrub tundra | LiCor Li200S Kipp & Zonen CMP3-L (post 2006) | in: 4.8 out: 4.4 | in: 203 out: 59 | 1995-2014 hourly |
| | Forest | LiCor Li200S Kipp & Zonen CMP3-L (post-2007) | in: 20.43 out: 14.33 | in: 234 out: 16 | 1995-2014 hourly |
| Ground heat flux [Wm$^{-2}$] | Alpine | REBS HFT3 | -0.08 | 2.65 | 1994-2013 hourly |
| | Shrub tundra | REBS HFT3 | -0.11 | 1.04 | 1993-2013 hourly |
| | Forest | REBS HFT3 | -0.16 | -0.65 | 1994-2013 hourly |



**Table 3. Hydrological data measured in the basin including variables, climatological water year mean values, current sensors, and measurement height. Negative heights show below ground depths**

| Variable | Site | Sensor | Height/depth [m] | Mean | Record period |
|---|---|---|---|---|---|
| Snow water equivalent [mm] | Alpine | Mt. Rose sampler | - | 64 | 1993-2014 monthly |
| | Shrub tundra | Mt. Rose sampler | - | 90 | 1993-2014 monthly |
| | Forest | Mt. Rose sampler | - | 37 | 1993-2014 monthly |
| Snow depth –April [mm] | Alpine | CSI UDG01 UDG01 (post-2005) | 1.65 | 20 | 1993-2014 monthly |
| | Shrub tundra | CSI UDG01 SR-50 (post-2005) | 1.67 | 52 | 1993-2014 monthly |
| | Forest | CSI UDG01 SR-50 (post-2007) | 1.41 | 12 | 1993-2014 monthly |
| Soil moisture [$m^3 m^{-3}$] | Alpine | CSI CS615 TDR | -0.05, -0.15 | 0.12, 0.16 | 1997-2013 hourly |
| | Shrub tundra | CSI CS615 TDR | -0.05, -0.15, -0.30, -0.80 | 0.19, 0.20, 0.20, 0.18 | 1996-2013 hourly |
| | Forest | CSI CS615 TDR | -0.05, -0.15, -0.30, -0.80 | 0.07, 0.10, 0.11, 0.14 | 1997-2013 hourly |
| Soil temperature [°C] | Alpine | YSI 40328 | 0, -0.03, -0.05, -0.15 | -0.7, -1.2, -1.6, -2.7 | 1997-2013 hourly |
| | Shrub tundra | YSI 40328 | -0.05, -0.10, -0.15, -0.30, -0.80 | 1.7, 1.3, 1.3, 1.2, 1.9 | 1996-2013 hourly |
| | Forest | YSI 40328 | 0, -0.05, -0.15, -0.30, -0.80 | -1.2, -0.1, -0.2, 0.2, 0.2, 0.9 | 1997-2013 hourly |
| Groundwater level [m] | Forest | HOBO (2001-07) Solinst (2008-2014) baro pressure loggers | - | -16.45 | 2001-2014 hourly |
| Streamflow [$m^3 s^{-1}$] | Alaska Highway | Various Instrumentation | - | 0.775 | 1993-2014 hourly |
| | Coal Lake | Logger level | - | 0.336 | 1994-2012 hourly |
| | Upper Wolf Creek | Logger level | - | 0.075 | 1994-2011 hourly |
| | Granger Creek | Logger level | - | 0.098 | 1998-2012 hourly |





**Table 4. Equations used to fill the gaps and correct the erroneous values in precipitation, relative humidity, and net radiation in the three stations representing forest, shrub tundra, and alpine biomes in the Wolf Creek Research Basin.**

| Variable | Equation | | No. |
|---|---|---|---|
| Snowfall | $S_N^{sh} = 1.339\, S_S^{sh}$ | $R^2 = 0.998$ | (1) |
| Snowfall | $S_N^{sh} = 1.427\, S_N^{WH}$ | $R^2 = 0.997$ | (2) |
| Rainfall | $R_S^{sh} = 1.333\, R_N^{WH}$ | $R^2 = 0.988$ | (3) |
| Snowfall | $S_N^A = 0.926\, S_N^{sh}$ | $R^2 = 0.999$ | (4) |
| Rainfall | $R_S^{sh} = 1.0394\, R_G^A$ | $R^2 = 0.996$ | (5) |
| Rainfall | $R_G^{sh} = 1.0036\, R_S^{sh}$ | $R^2 = 0.998$ | (6) |
| Snowfall Catch Ratio | $CR = 100 - 2.02\, U - 0.387\, U^2$ | | (7) |
| Relative humidity | $RH_{Corrected} = a.\, RH_{raw}^2 + b.\, RH_{raw} + 1$ | | (8) |
| Net radiation | $Rn_{corrected} = \begin{pmatrix} 1.0627\, Rn_{obs} & Rn_{obs} > 0 \\ 1.0079\, Rn_{obs} & Rn_{obs} < 0 \end{pmatrix}$ | | (9) |
| Net radiation | $Rn_{corrected} = \dfrac{A}{1 - \frac{0.054 \times 0.11U}{0.054 + 0.11U}}\, .\, Rn_{obs}\ ,$ $\quad \begin{matrix} Rn_{obs} > 0 \\ Rn_{obs} < 0 \end{matrix} \quad \begin{matrix} A = 1.182 \\ A = 0.962 \end{matrix}$ | | (10) |
| Net radiation | $Rn_{corrected} = \big(1 + 0.021286(U - 5)\big).\, Rn_{obs}\ ,\qquad U > 5\ \mathrm{ms}^{-1}$ | | (11) |



**Table 5. Regression equations used to fill the gaps in meteorological variables in the three stations representing forest, shrub tundra, and alpine biomes in the Wolf Creek Research Basin.**

| Station Dependent | Station Independent | Regression equation | Air temperature [ºC] | Relative humidity [%] | Wind speed [m s⁻¹] |
|---|---|---|---|---|---|
| Forest | Shrub tundra | slope | 1.18 | 0.80 | 0.29 |
| | | intercept | 0.91 | 12.64 | 0.89 |
| | | $R^2$ | 0.89 | 0.55 | 0.22 |
| | Alpine | slope | 1.20 | 0.75 | 0.14 |
| | | intercept | 2.74 | 11.84 | 0.88 |
| | | $R^2$ | 0.84 | 0.48 | 0.16 |
| | Data available (%) | | 94 | 93 | 76 |
| Shrub tundra | Forest | slope | 0.75 | 0.69 | 0.77 |
| | | intercept | -0.85 | 24.21 | 0.75 |
| | | $R^2$ | 0.89 | 0.55 | 0.22 |
| | Alpine | slope | 1.02 | 0.87 | 0.35 |
| | | intercept | 1.59 | 5.25 | 0.59 |
| | | $R^2$ | 0.95 | 0.74 | 0.36 |
| | Data available (%) | | 89 | 90 | 82 |
| Alpine | Forest | slope | 0.69 | 0.64 | 1.12 |
| | | intercept | -2.36 | 32.75 | 2.12 |
| | | $R^2$ | 0.84 | 0.48 | 0.16 |
| | Shrub tundra | slope | 0.93 | 0.85 | 1.01 |
| | | intercept | -1.59 | 16.14 | 1.84 |
| | | $R^2$ | 0.95 | 0.74 | 0.36 |
| | Data available (%) | | 87 | 87 | 78 |

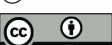



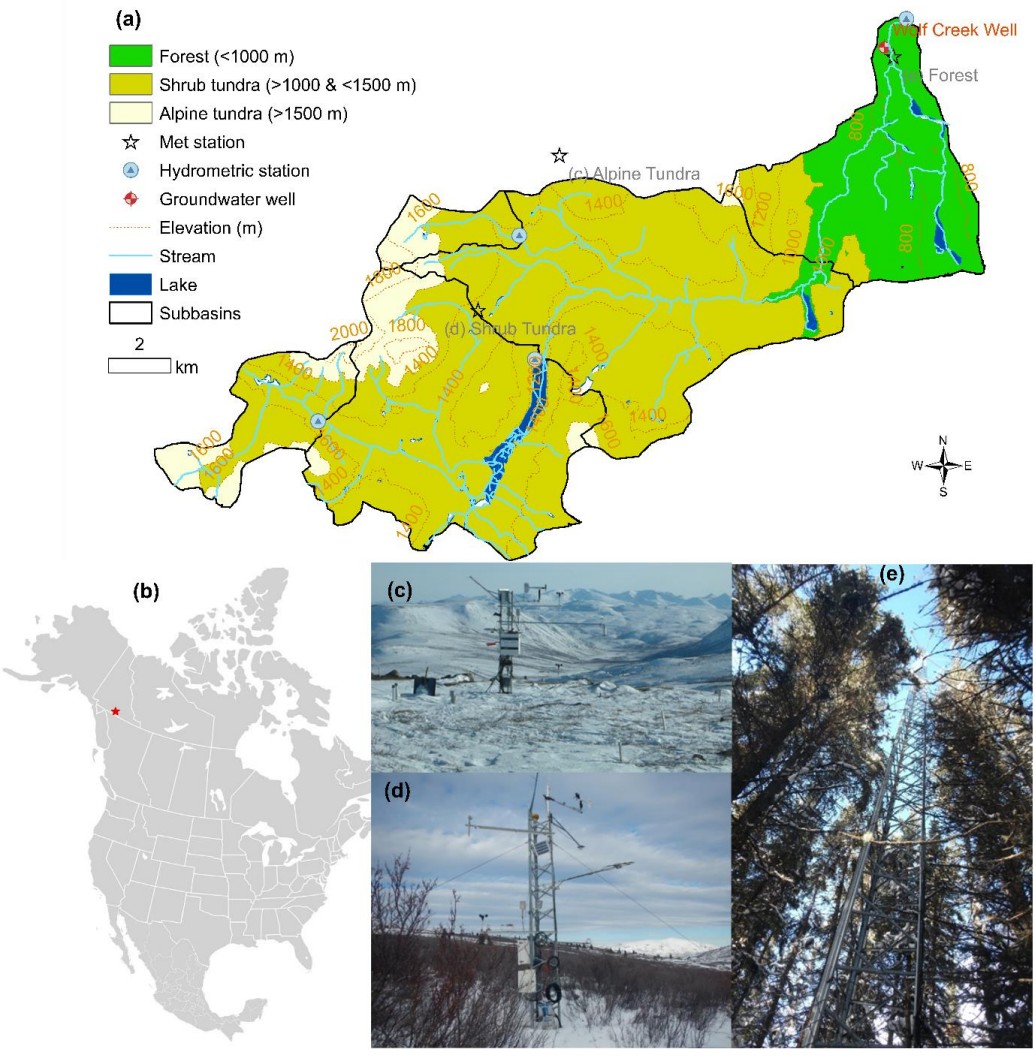

Figure 1. (a) The topography, streams, lakes, land-cover, groundwater well, and hydrometric stations, five sub-basins, and three meteorological stations within the Wolf Creek Research Basin, Yukon Territory, Canada shown in the (b) map of North America. Three meteorological stations represent (c) Alpine tundra, (d) Shrub tundra-taiga, and (e) forest biomes



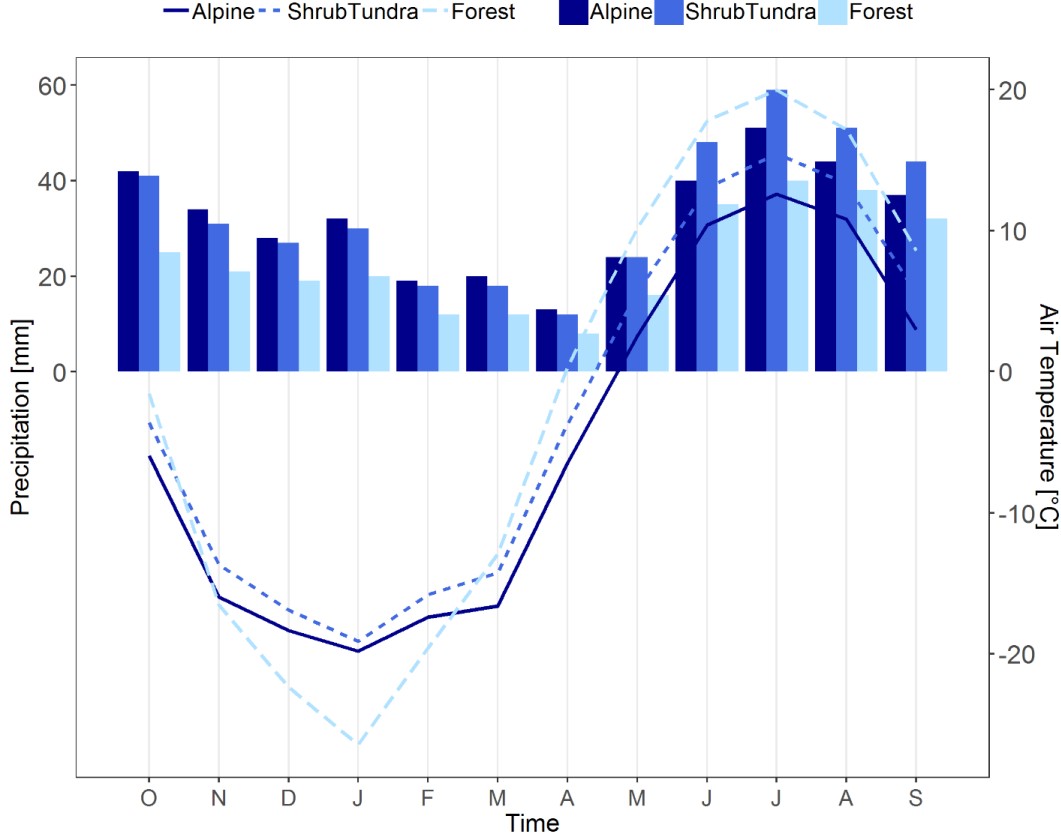

**Figure 2. Monthly precipitation and air temperatures in the three sites in Wolf Creek Research Basin averaged over the period of 1993-2014.**



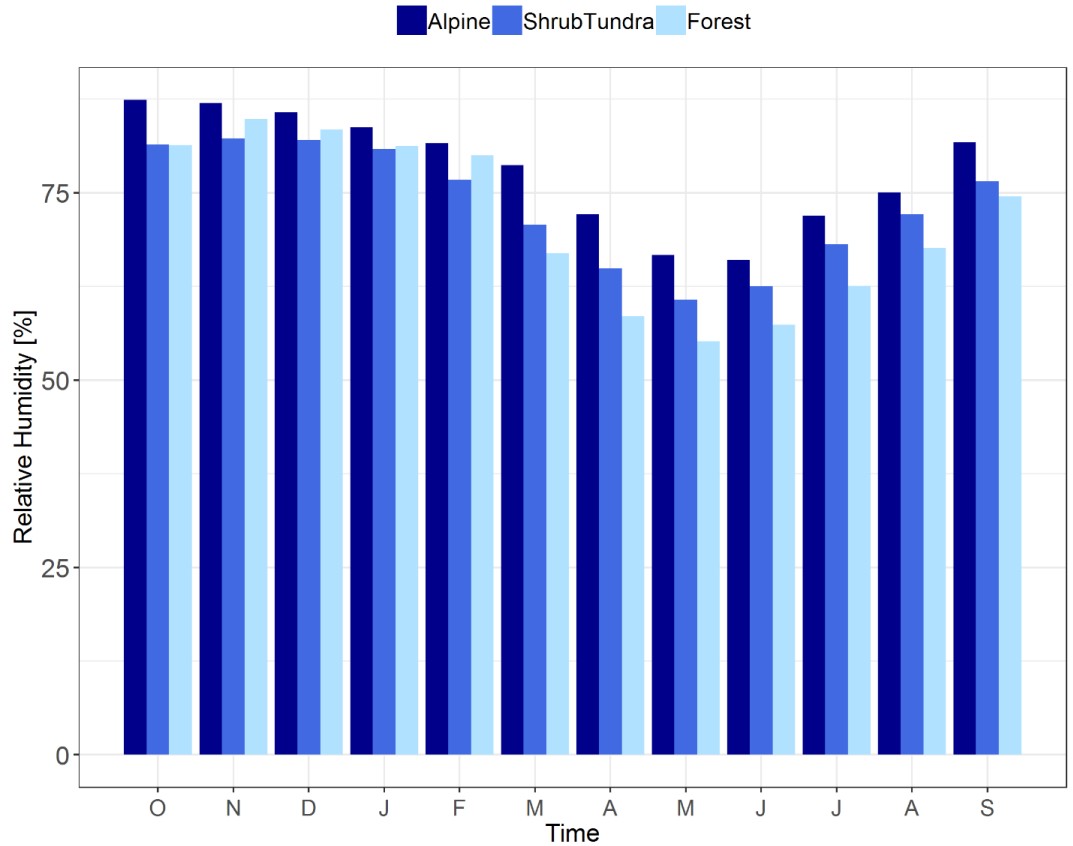

**Figure 3. Water year cycle of relative humidity in the three sites in Wolf Creek Research Basin averaged over the period of 1993-2014.**





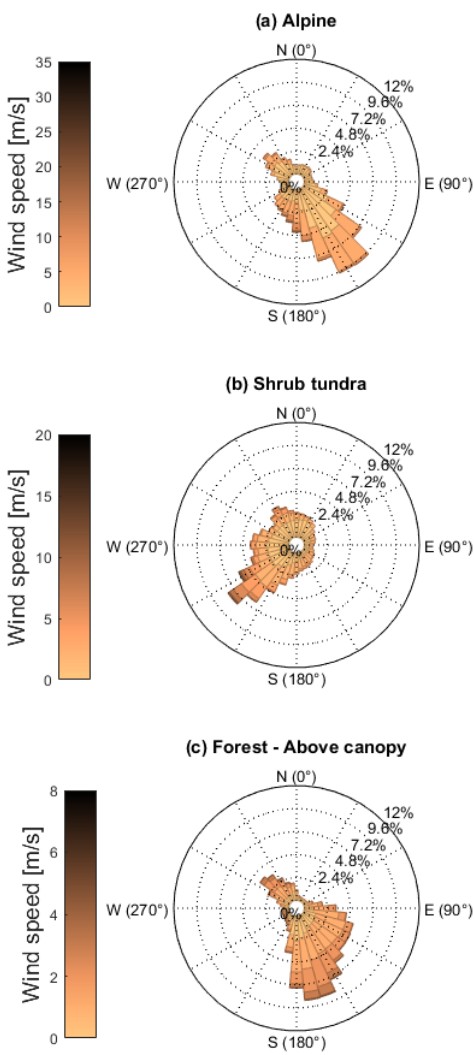

**Figure 4. Wind speed and direction in the three sites in Wolf Creek Research Basin averaged over the period of 1993-2014.**





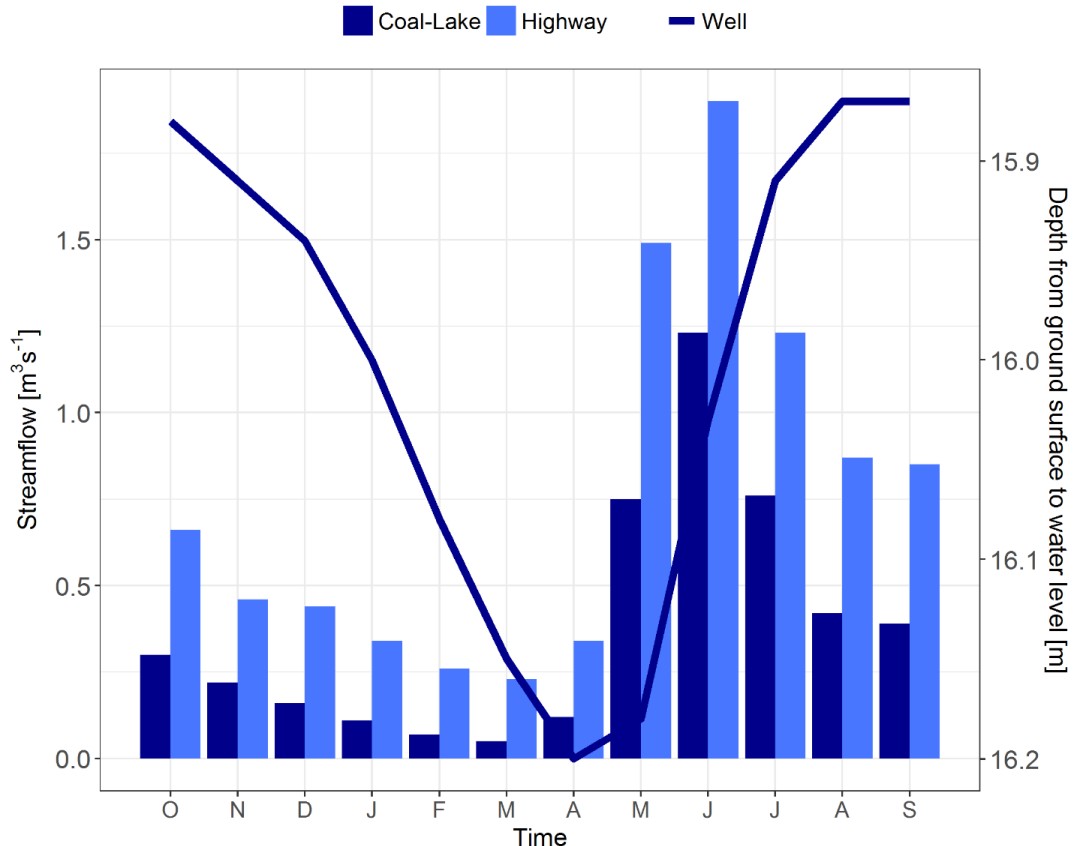

**Figure 5. Water year cycle of streamflow at two Alaska Highway and Coal lake gauges along with water levels in the groundwater well located in the forest biome within the Wolf Creek Research Basin.**




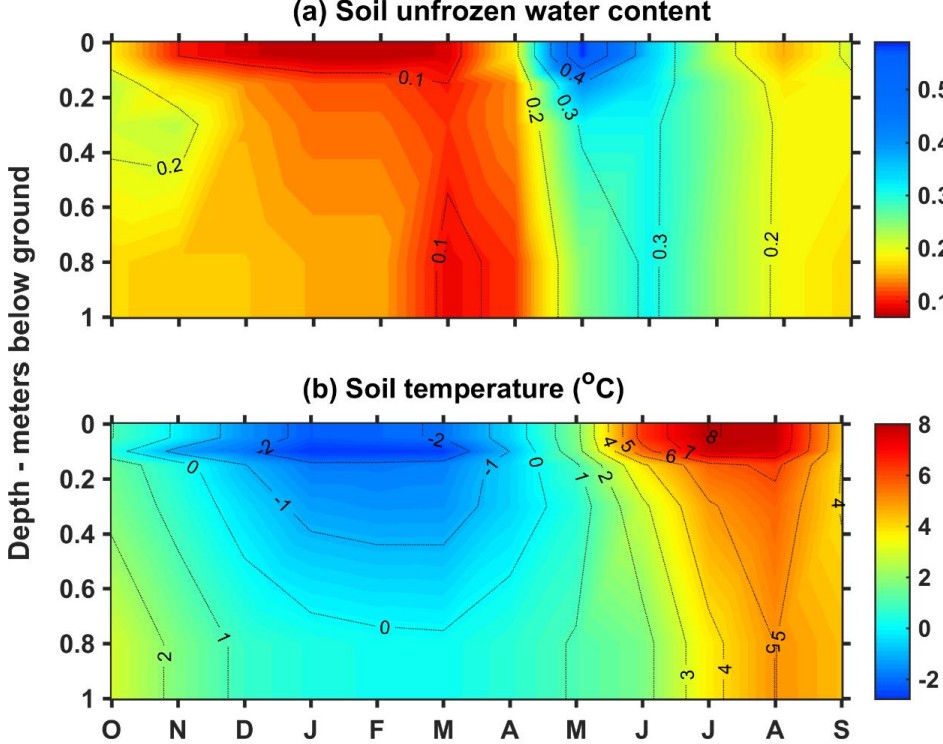

**Figure 6. Water year cycle of (a) the unfrozen water content of soil and (b) soil temperature at the Shrub tundra site in Wolf Creek Research Basin at depths of 0 to 1 m below ground surface.**

