# Peer review of "A long-term hydrometeorological dataset (1993-2014) of a northern mountain basin: Wolf Creek Research Basin, Yukon Territory, Canada"

_Earth System Science Data, 2018_

## Referee Comment (RC1) · Anonymous Referee #1 · 12 Nov 2018

Very few times I have considered to recommend publishing a paper as is. In this case, it can be done because the dataset is of outstanding quality in an environment where data scarcity is a common problem. The dataset has a great value hydrological, climatological (including validation of climate models) and ecological studies of a very cold environment. Wolf Creek is a very well know research basin and many previous publications have shown the extremely well equipped site (especially considering the tough conditions during winter). The paper is very well written, and the description of the data and how gaps were filled is very clear. The few limitations and uncertainties in

the data are also well reported. The access to the data is fast and intuitive. I have only found two small comments to consider when preparing the revised version. At some point is mentioned Ott pluvio and it should be OTT pluvio. When elevation is referred I would use m a.s.l. instead of m (elevation).

—————————————————————

---

## Referee Comment (RC2) · Anonymous Referee #2 · 17 Nov 2018

**GENERAL COMMENTS**

This manuscript submission describes a very unique long-term meteorological forcing and performance assessment dataset from the Wolf Creek Research Basin in Northern Canada. The authors present the dataset in a very simple and straightforward manner, while simultaneously demonstrating the effort required to prepare the data for use in a modeling or validation exercise.

The largest drawback of the dataset is that precipitation is only reported at daily time

scales. The rainfall ratio is given as 51% at the highest elevation site, and 61% at the lowest site (pg. 7, lines 28–30), meaning the WCRB lies well within the rain-snow transition elevation for its latitude. From an energy balance perspective, knowledge of when precipitation falls throughout a given day is crucial for determining precipitation phase. However, this drawback to the data does not have any bearing on the quality of this manuscript or affect the overall uniqueness of the dataset.

I recommend the manuscript be accepted after **minor revisions** have been addressed. These revisions are outlined in the following sections.

**SPECIFIC COMMENTS**

pg. 3, line 18 - How was the land cover image derived? Please describe.

pg. 3, line 22 - Just a suggestion: when referring to water years, "from 1993–1994 to 2013–2014" could be simplified to "water years 1994–2014". This occurs later in the text, as well. Also, the definition of a water year is never given in the text.

pg. 3, line 33 - The fact that precipitation is reported daily is mentioned only in passing. The reason(s) for not providing hourly precipitation should be described here.

pg. 4, line 12 - I would like to see a description of the snow survey methodology here at the end of this paragraph. (e.g. why are there 25 depth measurements to the 5 density measurements, and how are these measurements arranged?)

pg. 5, line 13 - Were the Standpipe gauge data corrected for undercatch similar to the Nipher gauge data?

pg. 5, line 18 - WSO is never defined.

pg. 5, line 22 - Rather than requiring the reader to go find that paper, how about providing a description of how phase was determined in just a few words and then citing the paper?

pg. 5, line 26 - This is an excellent way to fill gaps. Nice!

pg. 6, line 10 - How were these roughness lengths chosen? Citation?

pg. 6, line 31 - Should the REBS acronym be defined? I am unfamiliar with these instruments.

pg. 7, line 8 - The last sentence of this paragraph could be removed. Presently, it seems to be telling the reader how *they* should identify uncertainties in this dataset, but the reader did not collect the data. The responsibility of keeping detailed field notes is up to the data collectors. This sentence begs the reader to ask "If I want to identify uncertainties, were detailed field notes made and are they available?"

pg. 7, line 15 - You mention that albedo increases were used to help identify snowcovered periods. Were these albedo data derived from the shortwave radiometers? Were the outgoing shortwave measurements coherent with the incoming shortwave for all hours?

pg. 8, line 21 - Define the A2 scenario.

Table 4 - Provide the values of the $a$ and $b$ coefficients in Eq. (8).

**TECHNICAL CORRECTIONS**

pg. 2, line 13 - ...and has extensive parameter measurements.

pg. 2, line 19 - ...a long, cold snow season...

pg. 3, line 10 - Change "...pre- and post-move." to "...before and after the move."

pg. 3, line 15 - "A digital elevation model, or DEM, ..."

pg. 3, line 16 - "The DEM, ..."

pg. 4, line 7 - "...at the forest site and ..."

pg. 4, line 10 - "...snow depth, snow density, and snow water equivalent (SWE)..."

pg. 4, line 24 - Change "...it may be best..." to "...it is best..."

pg. 5, line 16 - "...wind undercatch are listed in Table 4."

pg. 5, line 32 - "...cumulative rainfall at the Standpipe gauge $(R_S^{sh})$ and rainfall at the Whitehorse WSO site..."

pg. 5, line 34 - "...the alpine site, a relationship..."

pg. 6, line 1 - "This equation..."

pg. 6, line 2 - "...the alpine site, the relationship..."

pg. 6, line 18 - "...after the gap was used."

pg. 6, line 33 - "Eq. (10) was used ..."

pg. 6, line 34 - "...wind speed, and if not available, cup anemometers..."

pg. 7, line 1 - "...when wind speed exceeded 5 ..."

pg. 7, line 29 - "...precipitation, decreases..."

pg. 8, line 13 - $8°C$ ?

pg. 8, line 14 - "Soil temperatures"

Table 2 caption - "...measured in three sites in the basin..."

---

## Author Comment (AC1) · 18 Dec 2018

Comments from Referee #1

RC: Very few times I have considered to recommend publishing a paper as is. In this case, it can be done because the dataset is of outstanding quality in an environment where data scarcity is a common problem. The dataset has a great value hydrological, climatological (including validation of climate models) and ecological studies of a very cold environment. Wolf Creek is a very well known research basin and many previous

publications have shown the extremely well equipped site (especially considering the tough conditions during winter). The paper is very well written, and the description of the data and how gaps were filled is very clear. The few limitations and uncertainties in the data are also well reported. The access to the data is fast and intuitive.

Response: We would like to thank Referee #1 for encouraging comments and sending us the reviews so quickly. We addressed the comments. We have provided all the changes in red color fonts in the revised manuscript.

RC: I have only found two small comments to consider when preparing the revised version. At some point is mentioned Ott pluvio and it should be OTT pluvio. When elevation is referred I would use m a.s.l. instead of m (elevation).

Response: We changed "Ott pluvio" to "OTT Pluvio" all over the text and in Table 2.

Response: We changed "m elevation" to "m a.s.l." in section 6.

\*\*\*\*\*\*\*\*\*\*\*\*\*\*\*\*\*\*\*\*\*\*\*\*\*\*\*\*\*\*\*\*\*\*\*\*\*\*\*\*\*\*\*\*\*\*\*\*\*\*\*\*\*\*\*\*\*\*\*\*\*\*\*\*\*\*\*\*\*\*\*\*\*\*\*\*\*\*\*\*\*\*\*\*\*

Comments from Referee #2

GENERAL COMMENTS

RC: This manuscript submission describes a very unique long-term meteorological forcing and performance assessment dataset from the Wolf Creek Research Basin in Northern Canada. The authors present the dataset in a very simple and straightforward manner, while simultaneously demonstrating the effort required to prepare the data for use in a modeling or validation exercise. The largest drawback of the dataset is that precipitation is only reported at daily time scales. The rainfall ratio is given as 51% at the highest elevation site, and 61% at the lowest site (pg. 7, lines 28–30), meaning the WCRB lies well within the rain-snow transition elevation for its latitude. From an energy balance perspective, knowledge of when precipitation falls throughout a given day is crucial for determining precipitation phase. However, this drawback to the data does not have any bearing on the quality of this manuscript or affect the overall uniqueness

of the dataset. I recommend the manuscript be accepted after minor revisions have been addressed.

Response: We would like to thank Referee #2 for his/her constructive comments and catching grammatical errors and typos. We addressed all of the comments as suggested. We agree that hourly precipitation data would be critical for separation of rain and snow. Hourly precipitation data have not been collected until recently that the Geonor T200B precipitation gauges were installed in the basin. In future, hourly data for precipitation will be uploaded to the database, which can be useful for determining precipitation phases and weather patterns. We will continuously update the database with recently recorded data.

These revisions are outlined in the following sections.

SPECIFIC COMMENTS

RC: pg. 3, line 18 - How was the land cover image derived? Please describe.

Response: "The land-cover classes were interpreted from Landsat 5 TM supervised image classification in August 1994 by National Hydrology Research Institute, Environment Canada." This is clarified in the manuscript.

RC: pg. 3, line 22 - Just a suggestion: when referring to water years, "from 1993–1994 to 2013–2014" could be simplified to "water years 1994–2014". This occurs later in the text, as well. Also, the definition of a water year is never given in the text.

Response: As suggested water years were simplified in the text and water year is defined as below in the first paragraph of section 4. In this paper it is assumed that water year starts on October 1st and ends on September 30th.

RC: pg. 3, line 33 - The fact that precipitation is reported daily is mentioned only in passing. The reason(s) for not providing hourly precipitation should be described here.

Response: The following sentence was added to the first paragraph of the section 4:

"Hourly precipitation data have not been collected until recently that the Geonor T200B precipitation gauges were installed in the basin. In future, hourly data for precipitation will be uploaded to the database, which can be useful for determining precipitation phases and weather patterns. We will continuously update the database with recently recorded data."

RC: pg. 4, line 12 - I would like to see a description of the snow survey methodology here at the end of this paragraph. (e.g. why are there 25 depth measurements to the 5 density measurements, and how are these measurements arranged?)

Response: This is now explained in the paper and the description is provided as below: "Each has a 25 point transect with for snow depth. Snow density is measured only at 5 out of 25 points5 density measurements. A relationship between snow depth and density was used to estimate density across the transect and then, measured and estimated snow density values were used to estimate SWE. Spatial variability of snow depth is higher than snow density and snow depth measurements are less time-consuming than density measurements, thus determining SWE with a small sample of density measurements and a large sample of depth measurements can represent average SWE across the transect more with reasonable accuracy."

RC: pg. 5, line 13 - Were the Standpipe gauge data corrected for undercatch similar to the Nipher gauge data?

Response: Since we have used a relationship between cumulative snowfall at the Standpipe and the Nipher gauges (Eq. 1 in Table 4) to calculate "Nipher equivalent" snowfall at the shrub tundra site and the "Nipher equivalent" snowfall data were corrected for undercatch, there is no need to correct Standpipe gauge data for undercatch.

RC: pg. 5, line 18 - WSO is never defined.

Response: "WSO" stands for "Weather Service Office" and in section 5 when this appears first in the text we now used "Whitehorse Weather Service Office (WSO) station"

[Figure]

instead of "Whitehorse WSO station".

RC: pg. 5, line 22 - Rather than requiring the reader to go find that paper, how about providing a description of how phase was determined in just a few words and then citing the paper?

Response: The following sentence added to clarify the method used for precipitation phase change: "First, total daily precipitation was separated to snowfall and rainfall phases using a method developed by Harder et al., (2013), in which psychrometric energy balance of the falling hydrometeors was calculated used to estimate precipitation phase based on the blowing snow sublimation turbulent transfer equations (Pomeroy and Gray, 1995)."

RC: pg. 5, line 26 - This is an excellent way to fill gaps. Nice!

Response: Thanks!

RC: pg. 6, line 10 - How were these roughness lengths chosen? Citation?

Response: The roughness lengths values were obtained from Pomeroy et al. (1997). This is now cited in the text and the following reference is added to the manuscript:

"Pomeroy, J. W., Marsh, P. and Gray, D. M.: Application of a distributed blowing snow model to the Arctic. Hydrological processes, 11(11), 1451–1464, 1997."

RC: pg. 6, line 31 - Should the REBS acronym be defined? I am unfamiliar with these instruments.

Response: REBS is defined as "Radiation and Energy Balance Systems (REBS)" in the manuscript.

RC: pg. 7, line 8 - The last sentence of this paragraph could be removed. Presently, it seems to be telling the reader how they should identify uncertainties in this dataset, but the reader did not collect the data. The responsibility of keeping detailed field notes is up to the data collectors. This sentence begs the reader to ask "If I want to identify

uncertainties, were detailed field notes made and are they available?"

Response: The following sentence is removed from the manuscript: "It is important to keep detailed field notes about icing to assist with identifying uncertainties with this data."

RC: pg. 7, line 15 - You mention that albedo increases were used to help identify snow-covered periods. Were these albedo data derived from the shortwave radiometers? Were the outgoing shortwave measurements coherent with the incoming shortwave for all hours?

Response: "Except for cases with erroneous data, outgoing and incoming shortwave radiations derived from the radiometers were temporally consistent, which were used to obtain changes in the albedo and snowcover." This is reflected in the manuscript now.

RC: pg. 8, line 21 - Define the A2 scenario.

Response: A2 scenario is clarified as "the A2 scenario of the Special Report on Emissions Scenarios (SRES)".

RC: Table 4 - Provide the values of the a and b coefficients in Eq. (8).

Response: The a and b coefficients in Eq. (8) were defined as below in Table 4 and RH_(raw, max) is defined in the text.

RH_Corrected=a. RH_raw^2+ b. RH_raw+1

a= 99/((RH_(raw, max) - 0.986).(RH_(raw, max)- 70))

b =0.986 − 70a

TECHNICAL CORRECTIONS

RC: pg. 2, line 13 - ...and has extensive parameter measurements.

Response: The word "has" is now added before "extensive parameter measurements."

RC: pg. 2, line 19 - ...a long, cold snow season...

Response: The word "and" instead of "," is now added before "cold snow season".

RC: pg. 3, line 10 - Change "...pre- and post-move." to "...before and after the move."

Response: The expression is changed as suggested.

RC: pg. 3, line 15 - "A digital elevation model, or DEM, ..."

Response: The word "DEM" is replaced by "or DEM,".

RC: pg. 3, line 16 - "The DEM, ..."

Response: The word "the" is now added before "DEM".

RC: pg. 4, line 7 - "...at the forest site and ..."

Response: The word "site" is now added after "at the forest".

RC: pg. 4, line 10 - "...snow depth, snow density, and snow water equivalent (SWE)..."

Response: The "snow depth, density, snow water equivalent (SWE)" is corrected as suggested.

RC: pg. 4, line 24 - Change "...it may be best..." to "...it is best..."

Response: Corrected as suggested.

RC: pg. 5, line 16 - "...wind undercatch are listed in Table 4."

Response: "were" is changed to "are".

RC: pg. 5, line 32 - "...cumulative rainfall at the Standpipe gauge (R_Sˆsh) and rainfall at the Whitehorse WSO site..."

Response: "The" is added before gauge and station names.

RC: pg. 5, line 34 - "...the alpine site, a relationship..."

Response: The symbol "," is now added.

RC: pg. 6, line 1 - "This equation..."

Response: "s" is removed from "equations".

RC: pg. 6, line 2 - "...the alpine site, the relationship..."

Response: The symbol "," is now added.

RC: pg. 6, line 18 - "...after the gap was used."

Response: The word "is" is changed to "was".

RC: pg. 6, line 33 - "Eq. (10) was used ..."

Response: The word "is" is changed to "was".

RC: pg. 6, line 34 - "...wind speed, and if not available, cup anemometers..."

Response: The word "and" is added before "if not available, cup anemometers".

RC: pg. 7, line 1 - "...when wind speed exceeded 5 ..."

Response: The word "exceed" is changed to "exceeded".

RC: pg. 7, line 29 - "...precipitation, decreases..."

Response: The symbol "," is now added.

RC: pg. 8, line 13 – 8oC ?

Response: The symbol "oC" is now added.

RC: pg. 8, line 14 - "Soil temperatures"

Response: "s" is add after "temperature".

RC: Table 2 caption - "...measured in three sites in the basin..."

Response: "s" is add after "site".

---

## Author Response (AR1)

**essd-2018-118**

**Response to comments on "A long-term hydrometeorological dataset (1993–2014) of a northern mountain basin: Wolf Creek Research Basin, Yukon Territory, Canada"**

By Kabir Rasouli et al.

**Comments from Referee #1**

Very few times I have considered to recommend publishing a paper as is. In this case, it can be done because the dataset is of outstanding quality in an environment where data scarcity is a common problem. The dataset has a great value hydrological, climatological (including validation of climate models) and ecological studies of a very cold environment. Wolf Creek is a very well known research basin and many previous publications have shown the extremely well equipped site (especially considering the tough conditions during winter). The paper is very well written, and the description of the data and how gaps were filled is very clear. The few limitations and uncertainties in the data are also well reported. The access to the data is fast and intuitive.

*We would like to thank Referee #1 for encouraging comments and sending us the reviews so quickly. We addressed the comments. We have provided all the changes in red color fonts in the revised manuscript.*

I have only found two small comments to consider when preparing the revised version. At some point is mentioned Ott pluvio and it should be OTT pluvio. When elevation is referred I would use m a.s.l. instead of m (elevation).
*We changed "Ott pluvio" to "OTT Pluvio" all over the text and in Table 2.*
*We changed "m elevation" to "m a.s.l." in section 6.*

**Comments from Referee #2**

GENERAL COMMENTS

This manuscript submission describes a very unique long-term meteorological forcing and performance assessment dataset from the Wolf Creek Research Basin in Northern Canada. The authors present the dataset in a very simple and straightforward manner, while simultaneously demonstrating the effort required to prepare the data for use in a modeling or validation exercise.

The largest drawback of the dataset is that precipitation is only reported at daily time scales. The rainfall ratio is given as 51% at the highest elevation site, and 61% at the lowest site (pg. 7, lines 28–30), meaning the WCRB lies well within the rain-snow transition elevation for its latitude. From an energy balance perspective, knowledge of when precipitation falls throughout a given day is crucial for determining precipitation phase. However, this drawback to the data does not have any bearing on the quality of this manuscript or affect the overall uniqueness of the dataset.

I recommend the manuscript be accepted after minor revisions have been addressed.
*We would like to thank Referee #2 for his/her constructive comments and catching grammatical errors and typos. We addressed all of the comments as suggested. We agree that hourly precipitation data would be critical for separation of rain and snow. Hourly precipitation data have not been collected until recently that the Geonor T200B precipitation gauges were installed in the basin. In future, hourly data for precipitation will be uploaded to the database, which can be useful for determining precipitation phases and weather patterns. We will continuously update the database with recently recorded data.*

These revisions are outlined in the following sections.

SPECIFIC COMMENTS

pg. 3, line 18 - How was the land cover image derived? Please describe.

*"The land-cover classes were interpreted from Landsat 5 TM supervised image classification in August 1994 by National Hydrology Research Institute, Environment Canada." This is clarified in the manuscript.*

pg. 3, line 22 - Just a suggestion: when referring to water years, "from 1993–1994 to 2013–2014" could be simplified to "water years 1994–2014". This occurs later in the text, as well. Also, the definition of a water year is never given in the text.
*As suggested water years were simplified in the text and water year is defined as below in the first paragraph of section 4. In this paper it is assumed that water year starts on October 1$^{st}$ and ends on September 30$^{th}$.*

pg. 3, line 33 - The fact that precipitation is reported daily is mentioned only in passing. The reason(s) for not providing hourly precipitation should be described here.
*The following sentence was added to the first paragraph of section 4:*
*"Hourly precipitation data have not been collected until recently that the Geonor T200B precipitation gauges were installed in the basin. In future, hourly data for precipitation will be uploaded to the database, which can be useful for determining precipitation phases and weather patterns. We will continuously update the database with recently recorded data."*

pg. 4, line 12 - I would like to see a description of the snow survey methodology here at the end of this paragraph. (e.g. why are there 25 depth measurements to the 5 density measurements, and how are these measurements arranged?)
*This is now explained in the paper and the description is provided as below:*
*"Each has a 25 point transect with for snow depth. Snow density is measured only at 5 out of 25 points. A relationship between snow depth and snow density was used to estimate density across the transect and then, measured and estimated snow density values were used to estimate SWE. Spatial variability of snow depth is higher than snow density. In addition, snow depth measurements are less time-consuming than density measurements, thus determining SWE with a small sample of density measurements and a large sample of depth measurements can represent average SWE across the transect with more reasonable accuracy."*

pg. 5, line 13 - Were the Standpipe gauge data corrected for undercatch similar to the Nipher gauge data?
*Since we have used a relationship between cumulative snowfall at the Standpipe and the Nipher gauges (Eq. 1 in Table 4) to calculate "Nipher equivalent" snowfall at the shrub tundra site and the "Nipher equivalent" snowfall data were corrected for undercatch, there is no need to correct the Standpipe gauge data for undercatch.*

pg. 5, line 18 - WSO is never defined.
*"WSO" stands for "Weather Service Office" and in section 5 when this appears first in the text we now used "Whitehorse Weather Service Office (WSO) station" instead of "Whitehorse WSO station".*

pg. 5, line 22 - Rather than requiring the reader to go find that paper, how about providing a description of how phase was determined in just a few words and then citing the paper?
*The following sentence added to clarify the method used for precipitation phase change:*
*"First, total daily precipitation was separated to snowfall and rainfall phases using a method developed by Harder et al., (2013), in which psychrometric energy balance of the falling hydrometeors was calculated and used to estimate precipitation phase based on the blowing snow sublimation turbulent transfer equations (Pomeroy and Gray, 1995)."*

pg. 5, line 26 - This is an excellent way to fill gaps. Nice!
*Thanks!*

pg. 6, line 10 - How were these roughness lengths chosen? Citation?
***The roughness lengths values were obtained from Pomeroy et al. (1997). This is now cited in the text and the following reference is added to the manuscript:***
***"Pomeroy, J. W., Marsh, P. and Gray, D. M.: Application of a distributed blowing snow model to the Arctic. Hydrological processes, 11(11), 1451–1464, 1997."***

pg. 6, line 31 - Should the REBS acronym be defined? I am unfamiliar with these instruments.
***REBS is defined as "Radiation and Energy Balance Systems (REBS)" in the manuscript.***

pg. 7, line 8 - The last sentence of this paragraph could be removed. Presently, it seems to be telling the reader how they should identify uncertainties in this dataset, but the reader did not collect the data. The responsibility of keeping detailed field notes is up to the data collectors. This sentence begs the reader to ask "If I want to identify uncertainties, were detailed field notes made and are they available?"
***The following sentence is removed from the manuscript: "It is important to keep detailed field notes about icing to assist with identifying uncertainties with this data."***

pg. 7, line 15 - You mention that albedo increases were used to help identify snowcovered periods. Were these albedo data derived from the shortwave radiometers? Were the outgoing shortwave measurements coherent with the incoming shortwave for all hours?
***"Except for cases with erroneous data, outgoing and incoming shortwave radiations derived from the radiometers were temporally consistent, which were used to obtain changes in the albedo and snowcover." This is reflected in the manuscript now.***

pg. 8, line 21 - Define the A2 scenario.
***The A2 scenario is clarified as "the A2 scenario of the Special Report on Emissions Scenarios (SRES)".***

Table 4 - Provide the values of the a and b coefficients in Eq. (8).
***The a and b coefficients in Eq. (8) were defined as below in Table 4 and $RH_{raw,\ max}$ is defined in the text.***

$$RH_{Corrected} = a.\ RH_{raw}^2 + b.\ RH_{raw} + 1$$
$$a = \frac{99}{(RH_{raw,\ max} - 0.986).(RH_{raw,\ max} - 70)}$$
$$b = 0.986 - 70a$$

TECHNICAL CORRECTIONS
pg. 2, line 13 - ...and has extensive parameter measurements.
***The word "has" is now added before "extensive parameter measurements."***

pg. 2, line 19 - ...a long, cold snow season...
***The word "and" instead of "," is now added before "cold snow season".***

pg. 3, line 10 - Change "...pre- and post-move." to "...before and after the move."
***The expression is changed as suggested.***

pg. 3, line 15 - "A digital elevation model, or DEM, ..."
***The word "DEM" is replaced by "or DEM,".***

pg. 3, line 16 - "The DEM, ..."
***The word "the" is now added before "DEM".***

pg. 4, line 7 - "...at the forest site and ..."
*The word "site" is now added after "at the forest".*

pg. 4, line 10 - "...snow depth, snow density, and snow water equivalent (SWE)..."
*The "snow depth, density, snow water equivalent (SWE)" is corrected as suggested.*

pg. 4, line 24 - Change "...it may be best..." to "...it is best..."
*Corrected as suggested.*

pg. 5, line 16 - "...wind undercatch are listed in Table 4."
*"were" is changed to "are".*

pg. 5, line 32 - "...cumulative rainfall at the Standpipe gauge ($R_s^{sh}$) and rainfall at the Whitehorse WSO site..."
*"The" is added before gauge and station names.*

pg. 5, line 34 - "...the alpine site, a relationship..."
*The symbol "," is now added.*

pg. 6, line 1 - "This equation..."
*"s" is removed from "equations".*

pg. 6, line 2 - "...the alpine site, the relationship..."
*The symbol "," is now added.*

pg. 6, line 18 - "...after the gap was used."
*The word "is" is changed to "was".*

pg. 6, line 33 - "Eq. (10) was used ..."
*The word "is" is changed to "was".*

pg. 6, line 34 - "...wind speed, and if not available, cup anemometers..."
*The word "and" is added before "if not available, cup anemometers".*

pg. 7, line 1 - "...when wind speed exceeded 5 ..."
*The word "exceed" is changed to "exceeded".*

pg. 7, line 29 - "...precipitation, decreases..."
*The symbol "," is now added.*

pg. 8, line 13 – 8$^{\circ}$C ?
*The symbol "$^{\circ}$C" is now added.*

pg. 8, line 14 - "Soil temperatures"
*"s" is add after "temperature".*

Table 2 caption - "...measured in three sites in the basin..."
*"s" is add after "site".*